# MEDIC: Zero-shot Music Editing with Disentangled Inversion Control

## Abstract

Text-guided diffusion models make a paradigm shift in audio generation, facilitating the adaptability of source audio to conform to specific textual prompts. Recent works introduce inversion techniques, like DDIM inversion, to zero-shot editing, exploiting pretrained diffusion models for audio modification. Nonetheless, our investigation exposes that DDIM inversion suffers from an accumulation of errors across each diffusion step, undermining its efficacy. Moreover, existing editing methods fail to achieve effective complex non-rigid music editing while maintaining essential content preservation and high editing fidelity. To counteract these issues, we introduce the *Disentangled Inversion* technique to disentangle the diffusion process into triple branches, rectifying the deviated path of the source branch caused by DDIM inversion. In addition, we propose the *Harmonized Attention Control* framework, which unifies the mutual self-attention control and cross-attention control with an intermediate Harmonic Branch to progressively achieve the desired harmonic and melodic information in the target music. Collectively, these innovations comprise the *Disentangled Inversion Control (DIC)* framework, enabling accurate music editing while safeguarding content integrity. To benchmark audio editing efficacy, we introduce *ZoME-Bench*, a comprehensive music editing benchmark hosting 1,100 samples spread across ten distinct editing categories. This facilitates both zero-shot and instruction-based music editing tasks. Our method achieves unparalleled performance in edit fidelity and essential content preservation, outperforming contemporary state-of-the-art inversion techniques.[1] Both code and benchmark will be released.

## 1 Introduction

Text-guided diffusion models (Song et al., 2020; Song & Ermon, 2020; Peebles & Xie, 2023) have made great progress in audio generation (Evans et al., 2024a;b), leveraging their impressive capability for realistic and varied outputs. In particular, these models (Liu et al., 2023b; Huang et al., 2023; Liu et al., 2024) provide the foundation for prompt-based audio editing, offering new opportunities to modify audio landscapes for specific *textual prompts*. Early audio editing strategies rely on training models from scratch (Copet et al., 2023; Agostinelli et al., 2023) or test-time optimization (Paissan et al., 2023; Plitsis et al., 2024), hampered by intensive computational demands. Recent works (Manor & Michaeli, 2024; Zhang et al., 2024) in zero-shot audio editing have been made through Denoising Diffusion Implicit Models (DDIM) (Song et al., 2020) and Denoising Diffusion Probabilistic Models (DDPM) (Ho et al., 2020) inversion techniques, but challenges remain.

Key among these challenges are maintaining *fidelity of editing* - ensuring the editing aligns with the provided instructions - and *essential content preservation*, ensuring that the unaltered musical attributes in the target prompts remain unchanged. However, balancing these objectives involves a careful exchange of information between the source and target branch in diffusion processes, with existing inversion methods like DDIM proving sub-optimal for conditional diffusion models (Mokady et al., 2023). Enhanced versions of edit-friendly DDPM inversion (Huberman-Spiegelglas et al., 2024) make strides in preservation by imprinting the source onto the noise space. However, this comes at the expense of reduced modification capabilities due to noise reduction.

---

[1] Audio samples are available at `https://MEDIC-Zero.github.io/`.

In this work, we rigorously examine the shortcomings of the DDIM inversion approach. Our comprehensive analysis indicates that while techniques like DDIM inversion provide an editable foundation for audio synthesis, they lack precision and may compromise the integrity of the original audio. The primary issue stems from the assumption of perfect reversibility in the ordinary differential equation (ODE) process, which is frequently not met during text-conditional editing. Consequently, this leads to distortions during the inversion. And the implementation of Classifier-free Guidance (CFG) (Ho & Salimans, 2021) aims to improve text adherence. However, it inadvertently amplifies the accumulated errors from the inversion process. Moreover, attention control (Cao et al., 2023; Hertz et al., 2022) has shown promise in achieving high fidelity and essential content preservation. For instance, MusicMagus (Zhang et al., 2024) introduces Cross-Attention Control for fine-grained music manipulation of rigid tasks. Nevertheless, these methods fail to resolve the issues of accumulated errors and struggle to achieve accurate editing for both rigid and non-rigid tasks, as illustrated in Figure 1. We introduce an innovative *Disentangled Inversion Control* technique to bridge this gap. This technique has two main components: *Harmonic Attention Control* and *Disentangled Inversion*. Cross-attention control (Hertz et al., 2022) and mutual self-attention control (Cao et al., 2023) have demonstrated robust editing capabilities for rigid and non-rigid image editing tasks, respectively. However, blindly combining these two approaches sequentially for music editing can result in sub-optimal outcomes, particularly in the original dual-branch setup, where it struggles with global attention refinement. To address this challenge, we introduce an intermediate branch called the Harmonic Branch, designed to modify both rigid and non-rigid attributes in music progressively. Furthermore, we disentangle the diffusion process into triple branches, correcting the deviation path caused by CFG in the source branch, which affects the essential content preservation. The other branches remain unchanged to ensure the highest possible edit fidelity.

Due to a lack of standardized benchmarks in audio editing, we introduce ZoME-Bench. The first music editing benchmark consists of 1,100 audio clips, distributing them into 10 rigorously curated editing categories across rigid and non-rigid tasks. Each entry is carefully assembled, comprising a source prompt, a target prompt, human instruction, and blended words intended for editing. Experimental results on ZoME-Bench indicate that MEDIC outperforms baselines, achieving significant improvements in essential content preservation and editing fidelity. Additionally, MEDIC demonstrates state-of-the-art performance in the variable-length music editing settings of the MusicDelta dataset.

Our contributions can be summarized as follows. 1) We introduce a novel, training-free methodology called Disentangled Inversion Control (DIC), designed to facilitate consistent manipulations of musical elements and intricate non-rigid editing tasks. 2) An *Harmonized Attention Control* framework is introduced to unify cross-attention and mutual self-attention control, which enables both rigid and non-rigid editing. 3) We present *Disentangled Inversion Technique* to achieve superior results with negligible inversion error by branch disentanglement and correction, aiding in accurately editing the music while preserving the content information. 4) We build a new benchmark for music editing, named *ZoME-Bench*, which supports both zero-shot and instruction-based music editing.

## 2 RELATED WORKS

**Text-based Audio editing.** The objective of text-based audio editing (Paissan et al., 2023; Plitsis et al., 2024; Han et al., 2023a) is to utilize diffusion models to manipulate audio content based on the provided target prompt. Existing methodologies for addressing these intricate challenges typically follow one of three paths. The first involves attempts to develop end-to-end editing models (Copet et al., 2023; Agostinelli et al., 2023; Chen et al., 2024)that employ diffusion processes. However, these efforts are often hampered by indirect training strategies or a lack of comprehensive datasets. The second path involves test-time optimization strategies that utilize large pre-trained models for editing (Paissan et al., 2023; Plitsis et al., 2024). Despite their versatility, these methods are often burdened by the significant computational demands of fine-tuning diffusion models or optimizing text-embeddings for signal reconstruction. Some methods may choose to employ both strategies (Kawar et al., 2023), further increasing the computational load. The final path involves inversion techniques, which typically use DDPM (Huberman-Spiegelglas et al., 2024; Wu & De la Torre, 2023)/DDIM (Song et al., 2020; Zhang et al., 2024) inversion strategies to extract diffusion noise vectors that match the source signal. Given its rapid and intuitive zero-shot editing capabilities, we have chosen inversion techniques as our primary research framework. In this work, we propose

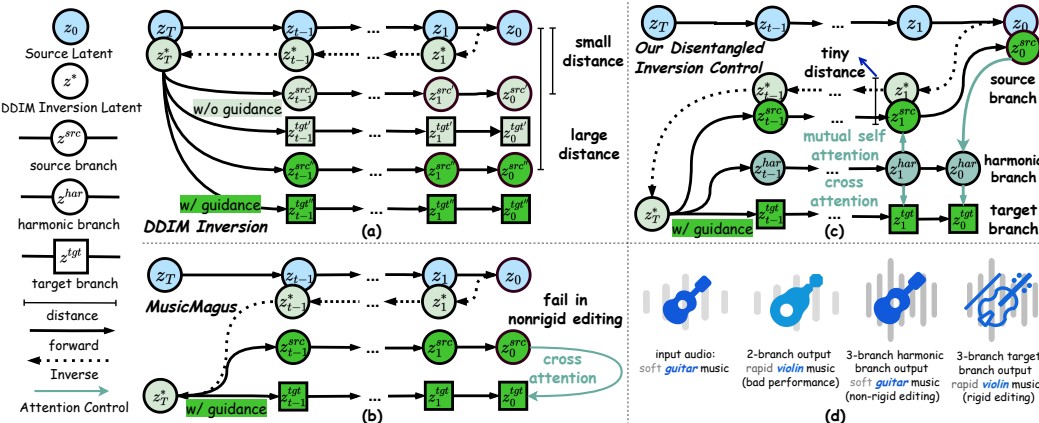

Figure 1: Comparisons of our method with two-branch inversion techniques, including DDIM Inversion and MusicMagus. (a) Framework of DDIM Inversion, showing configurations with and without classifier-free guidance. (b) Framework of MusicMagus, which incorporates cross-attention control. (c) Framework of our method, featuring disentangled inversion control. (d) An illustration comparing the output of the two-branch techniques with the progressive output of our triple-branch approach.

a new inversion technique named Disentangled Inversion Control. This technique aims to achieve accurate editing while preserving structural information.

**Inversion Techniques.** The field of image inversion techniques has experienced significant progress in recent years (Brooks et al., 2023; Kim et al., 2022; Parmar et al., 2023; Dhariwal & Nichol, 2021). While the DDIM inversion proves effective for unconditional diffusion models (Song et al., 2020; Wallace et al., 2023), its limitations become apparent when applied to text-guided diffusion models, particularly when classifier-free guidance is necessary for meaningful editing. A range of solutions (Mokady et al., 2023; Tumanyan et al., 2022) have been proposed to address these challenges. For example, Negative-Prompt Inversion strategically assigns conditioned text embeddings to Null-Text embeddings, effectively reducing potential deviation during editing. Conversely, Edit Friendly DDPM provides an alternative latent noise space via modified DDPM sample distributions, promoting the successful reconstruction of the desired image (Huberman-Spiegelglas et al., 2024). Optimization-based inversion methods using specific latent variables have recently gained popularity (Ju et al., 2023; Kawar et al., 2023). These are designed to minimize accumulated errors stemming from the DDIM inversion. Techniques such as Null-Text Inversion (Mokady et al., 2023) are promising, but they introduce complexity and instability into the optimization process, making it relatively time-consuming. Differently, we introduce a plug-in-plus method called Disentangled Inversion Control to separate branches, achieving superior performance with considerably fewer computational resources.

## 3 DISENTANGLED INVERSION CONTROL

### 3.1 PROBLEM DEFINITION AND BENCHMARK CONSTRUCTION

Despite significant work in text-to-audio generation models, particularly with the emergence of latent diffusion models (LDM), research on zero-shot music editing remains limited. Zero-shot music editing seeks to leverage the capabilities of text-to-music generation models to synthesize the desired music, denoted as $x_0^{tgt}$. This synthesized music should align with the target edited text prompt $\mathcal{P}^*$, which is directly modified from the source music $x_0^{src}$ and its corresponding text prompt $\mathcal{P}$. We compress source audio signals into latent $z_0^{src}$ for inversion.

To systematically validate our proposed method as a plug-and-play strategy for editing models, compare our method with existing zero-shot music editing methods, and compensate for the absence of standardized performance criteria for inversion and editing techniques, we construct a benchmark dataset named ZoME-Bench (Zero-shot Music Editing Benchmark). ZoME-Bench comprises 1,100 audio samples which are selected from MusicCaps (Doh et al., 2023), spanning ten different editing

types that include both rigid and non-rigid modifications. Each sample is accompanied by its corresponding source prompt, target prompt, human instruction, and source audio.

Additionally, we include annotations relevant to attention control, such as blended words, to facilitate detailed evaluations. Further details about our benchmark can be found in Appendix B.

## 3.2 MOTIVATION

Figure 1 and Preliminaries in Appendix A reveal that while techniques like DDIM inversion offer an editable base, they fall short of precision. This potentially compromising the essential content preservation. The implementation of Classifier-free Guidance (CFG) further amplifies the accumulated errors.

In the landscape of prompt-based editing (Dong et al., 2023; Kim et al., 2022; Feng et al., 2023), the ability to grasp the subtleties of linguistics and enable more granular cross-modal interactions stands as a formidable challenge. Hertz et al. (2022) acknowledges that in image editing, the fusion between text and visual modalities happens within the parameterized noise prediction network $\epsilon_\theta$. This leads to the development of various attention control techniques that guide the target denoiser network $\hat{\epsilon}_\theta$ in the image domain to better align with target prompts. Yet, analogous control mechanisms for non-rigid music editing are noticeably limited.

Taking these insights forward, we introduce Disentangled Inversion Control (DIC), a novel approach to achieve both rigid and non-rigid music editing. DIC strategically disentangles the diffusion process as triple branches, allowing each branch to optimize its functionality. At the same time, the strategy leverages harmonized attention control to facilitate targeted editing, thus aligning with the dual objectives of preserving the original audio essence and ensuring edit relevance. We will first introduce *Harmonized Attention Control* in Section 3.3 and discuss *Disentangled Inversion* in Section 3.4.

## 3.3 HARMONIZED ATTENTION CONTROL FRAMEWORK

The denoising architecture denoted as $\epsilon_\theta$, is structured as a sequence of fundamental blocks, each comprising a residual block (He et al., 2015) followed by self-attention and cross-attention modules (Vaswani et al., 2023; Dosovitskiy et al., 2020; Liu et al., 2023c). At the denoising step $t$, the output of the $(l-1)$-th block is passed through self-attention and then aligned with textual cues from prompt $P$ within the cross-attention layer. The attention mechanism is formalized as:

$$\text{Attention}(Q, K, V) = MV = \text{Softmax}\left(\frac{QK^T}{\sqrt{d}}\right)V \tag{1}$$

where $Q$ denotes the query features derived from the audio features, $K$ and $V$ represent the key and value features, and $M$ is the attention map. We explore varying semantic transformations of audio content through harmonized attention control strategies — cross-attention control for rigid changes and mutual self-attention control for non-rigid adjustments. Finally, we introduce an intermediate branch to host the desired harmonic and melodic information in the target music. The framework of harmonized attention control is depicted in Figure 2.

### 3.3.1 CROSS-ATTENTION CONTROL

Cross-attention Control (CAC) aims to inject the attention maps that are obtained from the generation with the original prompt $\mathcal{P}$, into a second generation with the target prompt $\mathcal{P}^*$. We follow the practice of Prompt-to-Prompt (Hertz et al., 2022) and define CAC as *Global Attention Refinement* and *Local Attention Blend*.

**Global Attention Refinement** At a given time step $t$, the attention map $M_t$ for both the origin and target branches is computed, averaging over all layers with respect to the noised latent $\mathbf{z}_t$. We employ an alignment function $A$ that maps each token index from the target prompt $\mathcal{P}^*$ to its equivalent in $\mathcal{P}$ or to None for non-aligning tokens. The refinement action is thus:

$$\text{Refine}(M_t^{src}, M_t^{tgt}, t) = \begin{cases} (M_t^{tgt})_{i,j} & \text{if } A(j) = \text{None}, \\ (M_t)_{i,j} & \text{otherwise}. \end{cases} \tag{2}$$

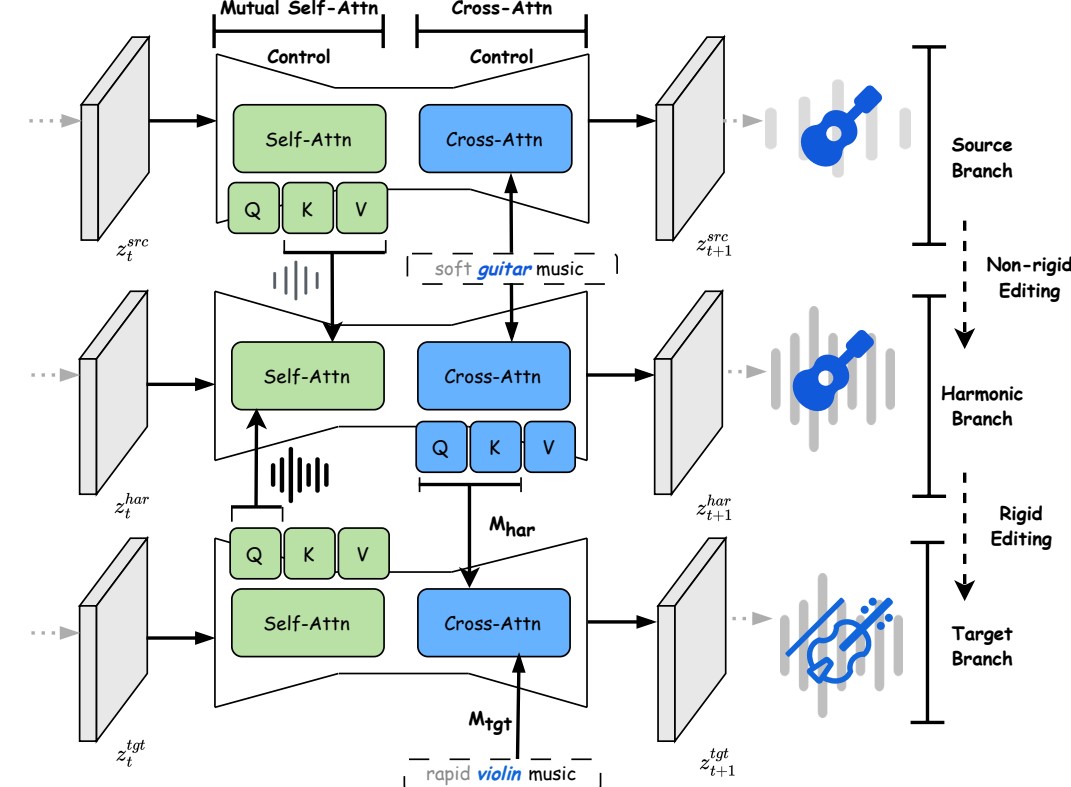

Figure 2: The framework of Harmonized Attention Control (HAC). HAC unifies cross-attention control and mutual self-attention control with an additional branch named *Harmonic Branch* to host the desired composition and structural information in the target music.

**Local Attention Blends** Beyond global attention, we incorporate a blending mechanism as suggested by Hertz et al. (2022) and Mokady et al. (2023). This method selectively integrates and maintains certain semantics by using target blend words $w^{tgt}$ for semantic additions and source blend words $w^{src}$ for semantic preservation. At each denoising step $t$, the mechanism operates on the target latent $\boldsymbol{z}_t^{tgt}$ as follows:

$$m_{\text{tgt}} = \text{Threshold}\left[M_t^{tgt}, w_{\text{tgt}}, k_{\text{tgt}}\right], \tag{3}$$

$$m_{\text{src}} = \text{Threshold}\left[M_t^{src}, w_{\text{src}}, k_{\text{src}}\right], \tag{4}$$

$$\boldsymbol{z}_t^{tgt} = (1 - m^{tgt} + m^{src}) \odot \boldsymbol{z}_t^{src} + (m^{tgt} - m^{src}) \odot \boldsymbol{z}_t^{tgt} \tag{5}$$

where $m_{\text{tgt}}$ and $m_{\text{src}}$ are binary masks and threshold function is as delineated below:

$$\text{Threshold}(M, k) = \begin{cases} 1 & \text{if } M_{i,j} \geq k, \\ 0 & \text{if } M_{i,j} < k. \end{cases} \tag{6}$$

For simplicity, we define the process of local editing in Equation 5 as:

$$\boldsymbol{z}_t^{tgt} = \text{LocalEdit}(\boldsymbol{z}_t^{src}, \boldsymbol{z}_t^{tgt}, M_t^{src}, M_t^{tgt}, w_{\text{src}}, w_{\text{tgt}}) \tag{7}$$

**Scheduling Cross-Attention Control** Applying cross-attention control only at early stages ensures creative flexibility while maintaining structure. Acting on insights from Hertz et al. (2022), we limit cross-attention to the initial phases up to a cutoff point $\tau_c$. This moderation allows us to capture the nuances and intended changes in musical compositions effectively. The approach is defined as:

$$\text{CrossEdit}(M^{src}, M^{tgt}, t) = \begin{cases} \text{Refine}(M_t^{src}, M_t^{tgt}) & \text{if } t \geq \tau_c, \\ M_t^{tgt} & \text{if } t < \tau_c. \end{cases} \tag{8}$$

---

**Algorithm 1** Harmonized Attention Control in one DDIM Forward Process

---

1: **Input:** A source prompt $\mathcal{P}$, a target prompt $\mathcal{P}^*$, a source audio latent $z_0$, denoising network $\epsilon_\theta(\cdot, \cdot, \cdot)$, current time step $\tau$, source and target blend words $w_{src}, w_{tgt}$, input latents $z_\tau^{src}, z_\tau^{tgt}, z_\tau^{har}$.

2: $\epsilon_{src}, \{Q^{src}, K^{src}, V^{src}\}, M_{src} = \epsilon_\theta(z_\tau^{src}, \tau, c_{src})$

3: $\epsilon_{tgt}, \{Q^{tgt}, K^{tgt}, V^{tgt}\}, M_{tgt} = \epsilon_\theta(z_\tau^{tgt}, \tau, c_{tgt})$

4: $\{Q^{har}, K^{har}, V^{har}\} = \text{SelfEdit}(\{Q^{src}, K^{src}, V^{src}\}, \{Q^{tgt}, K^{tgt}, V^{tgt}\}, \tau)$

5: $\epsilon_{har}, M_{har} = \epsilon_\theta(z_\tau^{har}, \tau, c_{src}; \{Q^{har}, K^{har}, V^{har}\})$

6: $\hat{M}^{tgt} = \text{CrossEdit}(M_{har}, M_{tgt}, \tau)$

7: $\hat{\epsilon_{tgt}} = \epsilon_\theta(z_\tau^{tgt}, \tau, c_{tgt}; \hat{M}^{tgt})$

8: $z_{\tau-1}^{src}, z_{\tau-1}^{tgt}, z_{\tau-1}^{har} = \text{Sample}([z_\tau^{src}, z_\tau^{tgt}, z_\tau^{har}], [\epsilon_{src}, \epsilon_{c_{tgt}}, \epsilon_{har}], \tau)$

9: $\boldsymbol{z}_{\tau-1}^{tgt} = \text{LocalEdit}(\boldsymbol{z}_{\tau-1}^{src}, \boldsymbol{z}_{\tau-1}^{tgt}, M_{\tau-1}^{src}, M_t^{tgt}, w_{src}, w_{tgt})$

10: **Output:** $z_{\tau-1}^{src}, z_{\tau-1}^{tgt}, z_{\tau-1}^{har}$

---

### 3.3.2 MUTUAL SELF-ATTENTION CONTROL

We diverge from the conventional use of cross-attention mechanisms and instead draw inspiration from the MasaCtrl (Cao et al., 2023) technique to refine music structure through self-attention queries. These queries adeptly navigate through non-rigid musical transformations, aligning with the designated musical theme or instrument (target prompt). The process begins by sketching the foundational musical theme using the target's self-attention components—$Q^{tgt}$, $K^{tgt}$, and $V^{tgt}$. This is followed by enriching this theme with elements resembling the thematic content from the source ($K^{src}$, $V^{src}$), steered by $Q^{tgt}$. However, applying this attentive modulation uniformly over all processing layers and through every denoising step might result in a composition excessively mirroring the source. Consequently, echoing the ethos of MasaCtrl, our proposed solution selectively employs mutual self-attention in the decoder portion of our music editing U-Net, initiated after a set number of denoising iterations.

**Scheduling Mutual Self-Attention Control**   The application of mutual self-attention is meticulously planned, beginning at a specific denoising step $S$ and extending beyond a designated layer $L$. The strength and influence of this control mechanism are designed as follows:

$$\text{SelfEdit}(Q^{src}, K^{src}, V^{src}, Q^{tgt}, K^{tgt}, V^{tgt}, t) = \begin{cases} Q^{src}, K^{src}, V^{src} & \text{if } t \geq S \text{ and } l \geq L, \\ Q^{tgt}, K^{src}, V^{src} & \text{otherwise} \end{cases} \quad (9)$$

In this framework, $S$ signifies the denoising step from which the mutual self-attention control commences, serving as a temporal threshold. Similarly, $L$ distinguishes the layer index below which this nuanced control strategy becomes operational, tailoring the musical output towards the intended artistic direction.

### 3.3.3 HARMONIC BRANCH INTEGRETION

The naive combination of cross-attention control and mutual self-attention control sequentially would lead to sub-optimal results in the original dual-branch setup, especially failing the global attention refinement. To address this issue, we introduce an additional latent harmonic branch, which serves as an intermediate to host the desired composition and structural information in the target music.

Our unified framework is detailed in Algorithm 2. During each forward step of the diffusion process, we start with mutual self-attention control on $z^{src}$ and $z^{tgt}$ and assign the output to the harmonic branch latent $z^{har}$. This lays the formal structure of the target music. Following this, cross-attention control is applied on $M^{har}$ and $M^{tgt}$ to refine the semantic information for $M^{tgt}$. As illustrated in Figure 2, the harmonic branch output $z_0^{har}$ reflects the requested non-rigid changes (e.g., "violin"), while preserving the rigid content semantics (e.g., "with noise"). The target branch output $z_0^{tgt}$ builds upon the structural layout of the $z^{har}$ while reflecting the requested rigid changes (e.g., "with noise").

---

**Algorithm 2** Disentangled Inversion Technique

---

1: **Input**: A source prompt $\mathcal{P}$, a target prompt $\mathcal{P}^*$, a source audio latent $\boldsymbol{z}_0$, and guidance scale $\omega$.
2: **Output**: A edited audio latent $\boldsymbol{z}_0^{tgt}$.
3: Compute the intermediate results $\boldsymbol{z}_T^*, ..., \boldsymbol{z}_1^*$ using DDIM inversion over $\boldsymbol{z}_0$.
4: Initialize $\boldsymbol{z}_T^{src} \leftarrow \boldsymbol{z}_T^*$, $\boldsymbol{z}_T^{tgt} \leftarrow \boldsymbol{z}_T^*$, $\boldsymbol{z}_T^{har} \leftarrow \boldsymbol{z}_T^*$.
5: **for** $t = T$ to 1 **do**
6: $\quad [\boldsymbol{d}_{t-1}^{src}, \boldsymbol{d}_{t-1}^{tgt}, \boldsymbol{d}_{t-1}^{har}] \leftarrow \boldsymbol{z}_{t-1}^*$ - DDIM_Forward($\boldsymbol{z}_t^{src}, t, [\mathcal{P}, \mathcal{P}^*, \mathcal{P}], \omega$)
7: $\quad \boldsymbol{z}_{t-1}^{src} \leftarrow$ DDIM_Forward($\boldsymbol{z}_t^{src}, t, [\mathcal{P}, \mathcal{P}^*, \mathcal{P}], \omega$) + $[\boldsymbol{d}_{t-1}^{src}, \boldsymbol{0}, \boldsymbol{0}]$
8: $\quad \boldsymbol{z}_{t-1}^{har} \leftarrow$ DDIM_Forward($\boldsymbol{z}_t^{har}, t, [\mathcal{P}, \mathcal{P}^*, \mathcal{P}], \omega$) + $[\boldsymbol{d}_{t-1}^{src}, \boldsymbol{0}, \boldsymbol{0}]$
9: $\quad \boldsymbol{z}_{t-1}^{tgt} \leftarrow$ DDIM_Forward($\boldsymbol{z}_t^{tgt}, t, [\mathcal{P}, \mathcal{P}^*, \mathcal{P}], \omega$) + $[\boldsymbol{d}_{t-1}^{src}, \boldsymbol{0}, \boldsymbol{0}]$
10: **end for**
11: **return** $\boldsymbol{z}_0^{tgt}$

---

### 3.4 Disentangled Inversion Technique

Recognizing the limitations of using DDIM inversion without classifier-free guidance, we observe that it yields an easily modifiable but imprecise approximation of the original audio signal. Increasing the guidance scale enhances editability, but sacrifices reconstruction accuracy due to latent code deviation during editing.

In order to address this issue, our methodology, which we have termed the *Disentangled Inversion Technique* disentangles into three branches: the source, the harmonic, and the target branch. This decoupling is designed to unleash the capabilities of each branch separately. For the source branch, we implement a targeted correction mechanism. By reintegrating the distance $z_t^* - z_t^{src}$ into $z_t^{src}$, we directly mitigate the deviation of the pathway. This straightforward adjustment effectively rectifies the path and minimizes the accumulated errors introduced by both DDIM inversion and classifier-free guidance, thereby enhancing consistency in the reconstructed audio. On the other hand, the target branch and harmonic branch are left unmodified to fully leverage the innate capabilities of diffusion models in generating the desired target audio. The branches' untouched state ensures that the model's potential is utilized to its fullest extent, thereby ensuring the fidelity and integrity of the generated audio. We will further discuss this in the Section 4.2.2. The algorithm of Disentangled Inversion Technique has been outlined in Algorithm 2.

Typical diffusion-based editing (Han et al., 2023b; Miyake et al., 2023) involves two parts: an inversion process to get the diffusion space of the audio, and a forward process to perform editing on the diffusion space. Disentangled Inversion can be plug-and-played into the forward process and rectifies the deviation path step by step. Specifically, Disentangled Inversion first computes the difference between $z_{t-1}^*$ and $z_{t-1}^{src}$, then adds the difference back to $z_{t-1}^{src}$ in DDIM forward. We only add the difference of the source prompt in latent space and update $z_{t-1}^{src}$, which is the key to retaining the editability of the target prompt's latent space.

## 4 Experiments

**Implementation Details.** We infer different editing methods using the pre-trained AudioLDM 2 (Liu et al., 2023b) models with 200 inference steps. The setting of baselines is followed by Manor & Michaeli (2024) and is all evaluated in NVIDIA A800 GPU for a fair comparison. We evaluate all methods in ZoMo-Bench across all editing types with fixed length. And we use the MusicDelta subset of the MelodyDB (Bittner et al., 2014) dataset for variable length comparisons, comprised of 34 musical excerpts in varying styles and in lengths ranging from 20 seconds to 5 minutes. Details can be found in Appendix C.

**Evaluation Metrics.** Our models employ a comprehensive evaluation using both objective and subjective metrics to assess essential content preservation, text-audio alignment fidelity, and audio quality. Objective metrics include Structure Distance (SD) (Ju et al., 2023), CLAP (Contrastive Language-Audio Pretraining) Score (Elizalde et al., 2023), LPAPS (Learned Perceptual Audio Patch Similarity) (Iashin & Rahtu, 2021; Paissan et al., 2023), and FAD (Fréchet Audio Distance) (Kilgour et al., 2018). The CLAP Score evaluates how well the edited audio aligns with the target prompt,

| Method | Objective Metrics | | | | Subjective Metrics | |
|---|---|---|---|---|---|---|
| | $SD_{\times 10^3}\downarrow$ | LPAPS↓ | FAD↓ | CLAP Score↑ | MOS-Q↑ | MOS-P↑ |
| AudioLDM 2 | 23.86 | 0.21 | 10.36 | 0.58 | 73.48 | 70.12 |
| MusicGen | 23.39 | 0.21 | 6.63 | 0.59 | 75.46 | 71.28 |
| SDEdit | 25.87 | 0.22 | 12.18 | 0.40 | 69.38 | 66.23 |
| DDIM Inversion | 22.52 | 0.21 | 9.51 | 0.49 | 73.10 | 3.38 |
| MusicMagus | 16.23 | 0.19 | 5.15 | 0.55 | 75.12 | 74.34 |
| DDPM-Friendly | 18.30 | 0.19 | 5.16 | 0.54 | 75.27 | 73.86 |
| **MEDIC** | **11.97** | **0.15** | **2.49** | **0.61** | **79.81** | **77.29** |

Table 1: Comparison with baselines on ZoME-Bench with fixed length about structure, content preservation, and CLAP similarity.

| Method | Objective Metrics | | | | Subjective Metrics | |
|---|---|---|---|---|---|---|
| | $SD_{\times 10^3}\downarrow$ | LPAPS↓ | FAD↓ | CLAP Score↑ | MOS-Q↑ | MOS-P↑ |
| AudioLDM 2 | 24.40 | 0.22 | 7.07 | 0.44 | 66.37 | 64.28 |
| MusicGen | 27.71 | 0.23 | 7.70 | 0.46 | 67.41 | 63.76 |
| SDEdit | 28.12 | 0.24 | 13.21 | 0.24 | 62.80 | 62.18 |
| DDIM Inversion | 23.5 | 0.21 | 10.12 | 0.27 | 65.94 | 65.73 |
| MusicMagus | 25.6 | 0.22 | 7.13 | 0.43 | 67.45 | 67.12 |
| DDPM-Friendly | 21.53 | 0.23 | 6.68 | 0.30 | 66.34 | 67.28 |
| **MEDIC** | **19.5** | **0.20** | **6.58** | **0.51** | **71.62** | **70.18** |

Table 2: Comparison with baselines in variable length setting.

while Structure Distance, LPAPS, and FAD, which have been adapted from image domain metrics, measure the similarity between the edited and source audio. Specifically, LPAPS quantifies the consistency of the edited audio in relation to the source audio, and the FAD metric assesses the distance between two distributions of audio signals. The Structure Distance measures the structural similarity between the edited and source audio. For the subjective evaluation, we conduct crowdsourced human assessments using the Mean Opinion Score (MOS) to evaluate both editing fidelity (MOS-Q) and content preservation (MOS-P). We attach the details of all metrics in Appendix C.1.

## 4.1 ZERO-SHOT MUSIC EDITING RESULTS

We present a comparative study of our *Disentangled Inversion Control* (DIC) against several established music generation and editing baselines, including AudioLDM 2 (Liu et al., 2023b), Music-Gen (Copet et al., 2023), SDEdit (Liu et al., 2023a), DDIM Inversion Ho et al. (2020), MusicMagus (Zhang et al., 2024), and DDPM-Friendly (Manor & Michaeli, 2024). Utilizing the ZoMo-Bench test set we developed, we assess the quality of the generated audio samples, focusing on two key aspects: Content Preservation and Editing Fidelity. The results, summarized in Table 1, lead to the following conclusions: (1) Our method, MEDIC, significantly outperforms both generation-based and inversion-based models in terms of editing fidelity and audio similarity, demonstrating its effectiveness in addressing complex editing tasks. (2) Although DDPM-Friendly and MusicMagus show improvements in essential content preservation, they struggle to maintain high text-audio alignment compared to generation models. MEDIC effectively addresses accumulated errors, achieving high audio similarity and an impressive CLAP score, reflecting its superior editing fidelity. (3) MEDIC also excels in subjective metrics, achieving the best performance in audio quality and content preservation.

**Variable Length Comparisons** We further evaluate MEDIC against baseline methods in a variable length setting, with the results presented in Table 2. Our analysis leads to the following conclusions: (1) MEDIC outperforms all baselines across all metrics, demonstrating the effectiveness of our approach in variable-length scenarios. (2) Inversion-based baselines experience a significant degradation in CLAP score, while MEDIC maintains the highest CLAP score of 0.51 on the MusicDelta dataset.

**Fine-grained Comparisons on ZoME-Bench** To substantiate the robustness of our method, we delve into fine-grained comparisons across different types of editing. The FAD and CLAP scores are depicted in Figure 3. The insights gleaned from this analysis are as follows: (1) Across all editing

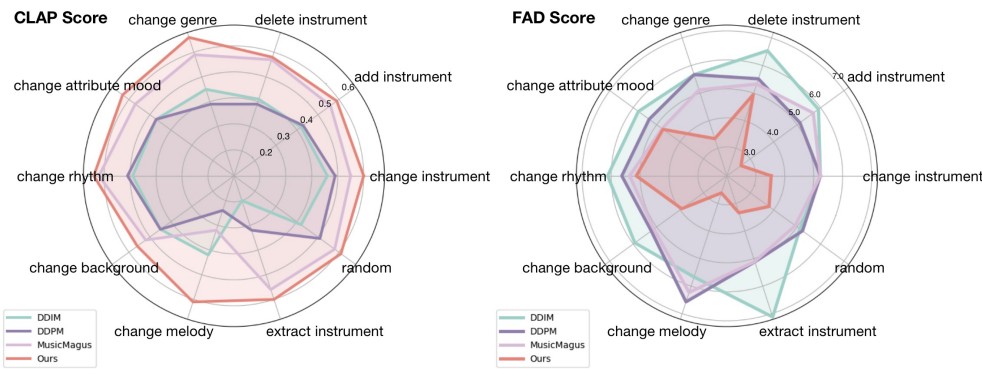

(a) Spider chart for CLAP score comparisons .   (b) Spider chart for FAD score comparisons.

Figure 3: A comprehensive performance evaluation on the ZoME-bench. We present spider charts of CLAP scores and FAD scores across 10 editing tasks for DDPM-Friendly, DDIM Inversion, MusicMagus, and Disentangled Inversion Control methods.

| Method | Structure Distance$_{\times 10^3}$ ↓ | LPAPS↓ | FAD↓ | CLAP Score↑ |
|---|---|---|---|---|
| **HAC** | **11.97** | **0.15** | **2.49** | **0.61** |
| w/o MSA Control | 14.13 | 0.19 | 2.75 | 0.56 |
| w/o CA Control | 13.78 | 0.18 | 2.50 | 0.58 |
| w/o Harmonic Branch | 12.75 | 0.16 | 2.59 | 0.59 |

Table 3: Ablation results about different control methods. MSA: Mutual Self-Attention, CA: Cross Attention.

types, our Disentangled Inversion Control surpasses other methods, demonstrating its prowess in handling both rigid and non-rigid editing tasks. (2) While baselines exhibit capabilities in handling certain rigid editing scenarios, they fall short in executing non-rigid manipulations, as reflected in their inferior performance, especially in the "Change Genre" and "Change Melody" non-rigid editing types.

## 4.2 ABLATION STUDY

The ablation studies presented in this section aim to validate the contributions of the Harmonized Attention Control and Disentangled Inversion to our framework's overall performance. Additionally, we examine the impact of the classifier-free guidance scale on editing outcomes, with a comprehensive analysis included in Appendix D.

### 4.2.1 ABLATION ON ATTENTION CONTROL METHODS

To validate the impact of our attention control mechanisms, we perform ablations on the following configurations: Remove Mutual Self Attention Control (w/o MSA Control), Remove Cross Attention Control (w/o CA Control), and without Harmonic Branch (w/o Harmonic Branch). The findings, detailed in Table 3, reveal that: (1) Both cross-attention and mutual self-attention controls individually enhance editing performance. (2) Although the naive combination of mutual self-attention control and cross-attention control improves the preservation and fidelity, it still yields sub-optimal outcomes due to the lack of a progressive harmonic branch. This demonstrates the effectiveness of harmonized attention control.

### 4.2.2 ABLATION ON DISENTANGLED INVERSION TECHNIQUE

To demonstrate the soundness of Algorithm 2 and the effective disentanglement of triple branches, we can draw conclusions from Table 4: 1) Incorporating source distance into the target latent and harmonic branch, this leads to a decline in both preservation metrics and CLAP similarity score. 2) Incorporating target distance into the target branch and harmonic branch improves structural

| Distance | Structure Distance$_{\times 10^3}$ ↓ | LPAPS↓ | FAD↓ | MSE$_{\times 10^5}$ ↓ | CLAP Score↑ |
|---|---|---|---|---|---|
| $[\boldsymbol{d}_{src}, \boldsymbol{d}_{src}, \boldsymbol{0}]$ | 14.28 | 0.17 | 2.64 | 4.67 | 0.57 |
| $[\boldsymbol{d}_{src}, \boldsymbol{0}, \boldsymbol{d}_{src}]$ | 13.17 | 0.17 | 2.55 | 4.59 | 0.58 |
| $[\boldsymbol{d}_{src}, \boldsymbol{d}_{tgt}, \boldsymbol{0}]$ | 11.24 | 0.15 | 2.44 | 4.32 | 0.56 |
| $[\boldsymbol{d}_{src}, \boldsymbol{0}, \boldsymbol{d}_{tgt}]$ | **11.12** | **0.14** | **2.41** | **4.29** | 0.56 |
| $[\boldsymbol{d}_{src}, \boldsymbol{d}_{har}, \boldsymbol{0}]$ | 37.51 | 0.27 | 27.2 | 14.23 | 0.28 |
| $[\boldsymbol{d}_{src}, \boldsymbol{0}, \boldsymbol{0}]$ | 11.97 | 0.15 | 2.49 | 4.54 | **0.61** |

Table 4: Ablation results from the disentangled inversion technique. $[\cdot, \cdot, \cdot]$ denotes adding the distance (line 6 in Algorithm 2). MSE means the mean square error loss between the edited audio features and source audio features.

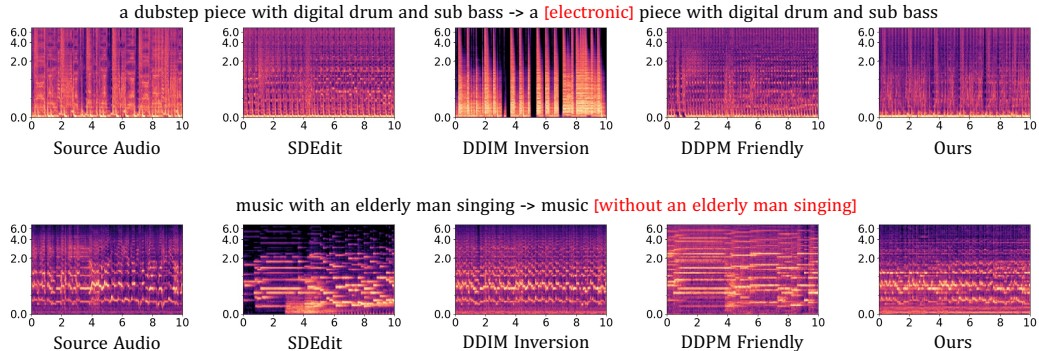

Figure 4: Visualizations of the source audio's mel and edited mel-spectrograms by different editing methods.

integrity and content preservation. However, this modification leads to a significant decrease in the CLAP score, indicating that while structural quality may improve, the alignment with the target prompt suffers. 3) Adding harmonic distance to the harmonic branch leads to a noticeable decline in performance. This finding showcases that introducing excessive deviations in the harmonic branch may adversely affect the overall audio quality and coherence.

### 4.3 QUALITATIVE RESULTS

To complement our quantitative findings, we present a qualitative comparison in Figure 4. Methods such as SDEdit and inversion-based techniques often struggle to balance high editability with preserving melodic content and harmonic structure. In contrast, our Disentangled Inversion Control performs better in precise music editing while preserving structural integrity. We provide additional qualitative results in the Appendix G for further examination.

## 5 CONCLUSION

In this paper, we explored the burgeoning realm of text-guided diffusion models for audio generation, recognizing their potential to reshape source audio in alignment with specific textual prompts. We proposed the Disentangled Inversion Control to support both rigid and non-rigid editing tasks. Instead of a two-branch setting, we add an intermediate branch named the harmonic branch to progressively integrate harmonic and melodic information in music by cross-attention control and mutual self-attention control. To counteract the accumulated errors caused by DDIM inversion and CFG, we introduced a simple but effective method named disentangled inversion to separate the diffusion process into triple branches and eliminate the latent discrepancy distance in the source branch. Our comprehensive evaluations, conducted on the *ZoME-Bench*—a robust benchmark for music editing comprising 1,100 samples across 10 varied editing categories—attested to the superiority of our methods. And the experiment results on variable length test sets and ablation studies further validated the effectiveness and robustness of our method. We envisage that our work could serve as a basis for future zero-shot music editing studies.

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

# A  PRELIMINARIES AND ANALYSES

This section introduces the foundational concepts of DDIM sampling and classifier-free guidance as applied to diffusion models for text-guided audio synthesis. It further explores the challenges associated with these methods.

## A.1  DIFFUSION MODELS

Text-guided diffusion models aim to map a random noise vector $z_t$ and textual condition $c$ to an output audio $z_0$, corresponding to the given conditioning prompt. We train a denoiser network $\epsilon_\theta(z_t, t, c)$ to predict the Gaussian noise $\epsilon \in \mathcal{N}(0, \mathbf{I})$ following the objective:

$$\min_\theta \mathbf{E}_{z_0, \epsilon \in \mathcal{N}(0, \mathbf{I}), t \in Uniform(1, T)} ||\epsilon - \epsilon_\theta(z_t, t, c)||^2 \tag{10}$$

Where noise is added to the sampled data $z_0$ according to timestamp $t$. At inference, given a noise vector $z_T$, the noise $i$ is gradually removed by sequentially predicting it using a pre-trained network for $T$ steps. To generate audios from given $z_T$, we employ the deterministic DDIM sampling:

$$z_{t-1} = \frac{\sqrt{\alpha_{t-1}}}{\sqrt{\alpha_t}} z_t + \sqrt{\alpha_{t-1}}(\sqrt{\frac{1}{\alpha_{t-1}} - 1} - \sqrt{\frac{1}{\alpha_t} - 1})\epsilon_\theta(z_t, t, c) \tag{11}$$

## A.2  DDIM INVERSION

While diffusion models have superior characteristics in the feature space that can support various down-stream tasks, it is hard to apply them to audios in the absence of natural diffusion feature space for non-generated audios. Thus, a simple inversion technique known as DDIM inversion is commonly used for unconditional diffusion models, predicated on the presumption that the ODE process can be reversed in the limit of infinitesimally small steps:

$$z_t^* = \frac{\sqrt{\alpha_t}}{\sqrt{\alpha_{t-1}}} z_{t-1}^* + \sqrt{\alpha_t}(\sqrt{\frac{1}{\alpha_t} - 1} - \sqrt{\frac{1}{\alpha_{t-1}} - 1})\epsilon_\theta(z_{t-1}^*, t - 1) \tag{12}$$

However, in most text-based diffusion models, this presumption cannot be guaranteed, resulting in a perturbation from $z_t$ to $z_t^*$. Consequently, an additional perturbation from $z_t^*$ to $z_t^{src}$ arises when sampling an audio from $z_T^*$ where $\alpha$ is hyper-parameter:

$$z_t = \sqrt{\alpha_t} z_0 + \sqrt{1 - \alpha_t}\epsilon \tag{13}$$

## A.3  CLASSIFIER-FREE GUIDANCE

Classifier-free Guidance (CFG) (Ho & Salimans, 2021) is proposed to overcome the limitation of text-conditioned models, where text adherence could be weak. However, a higher guidance scale, $\omega$, which is intended to strengthen the model's fidelity to the text prompt, inadvertently magnifies the accumulated inversion error. This becomes problematic in editing scenarios where precise control over the audio synthesis is desired. The modified noise estimation in CFG can be expressed as:

$$\hat{\epsilon}_\theta(z_t, t, c, \varnothing) = \omega \cdot \epsilon_\theta(z_t, t, c) + (1 - \omega) \cdot \epsilon_\theta(z_t, t, \varnothing) \tag{14}$$

where $\varnothing$ = ("") is the embedding of a null text. This further leads to another perturbation from $z_t'$ to $z''$ due to the destruction of the DDIM process and causes error augmentation as demonstrated in Figure 1.

# B  BENCHMARK CONSTRUCTION

## B.1  GENERAL INFORMATION

Here are the details of our ZoME-Bench dataset (Zero-shot Music Editing Benchmark). This dataset contains 1,000 audio samples, selected from MusicCaps, with each sample being 10 seconds long and having a sample rate of 16k.

We refactor the original captions to express specific edits and divide them into 10 parts, each representing a different type of editing. A sample and details are shown in the following table 5.

| Editing type id | Editing type | size | origin prompt | editing prompt | editing instruction |
|---|---|---|---|---|---|
| 0 | change instrument | 131 | ambient acoustic [guitar] music | ambient acoustic [violin] music | change the instrument from guitar to violin |
| 1 | add instrument | 139 | metal audio with a distortion guitar [and drums] | metal audio with a distortion guitar | add drums to the piece |
| 2 | delete instrument | 133 | an eerie tense instrumental featuring electronic drums [and synth keyboard] | an eerie tense instrumental featuring electronic drums | remove the synth keyboard |
| 3 | change genre | 134 | a recording of a solo electric guitar playing [blues] licks | a recording of a solo electric guitar playing [rocks] licks | change the genre from blues to rock |
| 4 | change mood | 100 | a recording featuring electric bass with an [upbeat] vibe | a recording featuring electric bass with an [melancholic] vibe | turn upbeat mood into melancholic mood |
| 5 | change rhythm | 69 | a live ukulele performance featuring [fast] strumming and emotional melodies | a live ukulele performance featuring [slow] strumming and emotional melodies | change fast rhythm into slow one |
| 6 | change background | 95 | female voices in unison with [acoustic] guitar | female voices in unison with [electric] guitar | switch acoustic guitar to electric guitar |
| 7 | change melody | 121 | this instrumental song features a [relaxing] melody | this instrumental song features a [cheerful] melody | change relaxing melody into cheerful melody |
| 8 | extract instrument | 111 | a reggae rhythm recording with bongos [djembe congas acoustic drums and electric guitar] | a reggae rhythm recording with bongos | extract bongos from the recording |
| 9 | random | 67 | / | / | / |

Table 5: Information of ZoME-Bench dataset

## B.2 ANNOTATION PROCESS

We rebuild our caption from captions for Musiccaps offered by (Agostinelli et al., 2023). With the help of ChatGPT-4 (OpenAI, 2023), we rebuild the caption with prompt as follows(take type "change melody" as examples):

**Description**: "There is a description of a Piece of music, Please judge whether the description has information of melody. If not, just answer "Flase", else change its melody properly into the opposite one, just change the adjective and don't replace any instrument! ","blended_word" is [origin melody,

changed melody], "emphasize" is [changed melody], "blended_word" and "emphasize" are tuples.

**Question**: (A mellow, passionate melody from a noisy electric guitar)

**Answer**:("source_prompt": "A mellow, [passionate] melody from a noisy electric guitar", "editing_prompt": "A mellow, [soft] melody from a noisy electric guitar", "blended_word": ["passionate melody", "soft melody"], "emphasize": ["soft melody"])

**Question**: (A recording of solo harp music with a dreamy, relaxing melody.)

**Answer**: ("source_prompt": "A recording of solo harp music with a dreamy, [relaxing] melody.", "editing_prompt": "A recording of solo harp music with a dreamy, [nervous] melody.", "blended_word": ["relaxing melody","nervous melody"], "emphasize" :["nervous melody"])

**Question**: ("A vintage, emotional song with mellow harmonized flute melody and soft wooden percussions")

**Answer**: ("source_prompt": "A vintage, emotional song with [passionate] flute melody and soft wooden percussions.", "editing_prompt": "A vintage, emotional song with [harmonized] flute melody and soft wooden percussions.", "blended_word": ["harmonized flute melody","passionate flute melody"]), "emphasize" :["passionate flute melody"])

**Now we have Question**:({origin caption}), Answer(?)"

In the same way, instructions are appended by prompt as follows (take type "change melody" as examples):

**Description**: "There are two descriptions of different pieces of music divided by &, Please describe the difference you need to give me the results in the following format: Question: this instrumental song features a [relaxing] melody with a country feel accompanied by a guitar piano simple percussion and bass in a slow tempo & this instrumental song features a [cheerful] melody with a country feel accompanied by a guitar piano simple percussion and bass in a slow tempo

**Answer**: change relaxing melody into cheerful melody

**Question**: this song features acapella harmonies with a [high pitched] melody complemented by both high pitched female whistle tones and male low pitch tones & this song features acapella harmonies with a [smooth] melody complemented by both high pitched female whistle tones and male low pitch tones

**Answer**: turn a high pitched melody into smooth melody

**Question**: a traditional and hopeful song with a harmonizing throaty male vocal and [dissonant] background melody from strings albeit presented in low quality & a traditional and hopeful song with a harmonizing throaty male vocal and [harmonic] background melody from strings albeit presented in low quality

**Answer**: change dissonant melody into harmonic melody

**Now we have Question**: ['source prompt'] & ['editing prompt'], Answer(?)"

Through this method, supplemented by rounds of manual review, we ensure the quality of this benchmark.

### B.3 DATA FORMAT

Taking the first piece as an example, we express our data in JSON format with six keys

```
{
        "000000000000": {
        "editing_prompt": "a live recording of ambient acoustic
        [violin] music",
        "source_prompt": "a live recording of ambient acoustic
        [guitar] music",
        "blended_word": "(\" guitar \", \" violin \")",
        "emphasize": "(\" violin \")",
        "audio_path": "wavs/MusicCaps_-4SYC2YgzL8.wav",
        "editing_type_id": "0",
        "editing_instruction": "change the instrument from guitar
        to violin"
    }
}
```

"Editing_prompt" refers to the edited caption, while "source_prompt" denotes the original caption. "Blended_word" indicates the subject to be edited, and "Emphasize" represents the word that should be highlighted. "Editing_instruction" provides a description of the editing process. Additionally, in the editing type "delete instrument," we introduce another key, "neg_prompt", which helps reduce the likelihood of deleted instruments reappearing.

## C    IMPLEMENTATION DETAILS

For our evaluation, we utilize the pre-trained AudioLDM 2-Music model (Liu et al., 2023b). Our assessment employs a comprehensive set of metrics, namely CLAP, LPAPS, Structure Distance, and FAD. These metrics are calculated using the CLAP models available in the AudioLDM_eval package, which is accessible at `https://github.com/haoheliu/audioldm_eval`. In line with the methodology described by Manor & Michaeli (2024), we apply a forward guidance of 3 and a reverse guidance scale of 12 for DDPM inversion. For the DDIM inversion, the guidance scale is set to 5, while for SDEdit, we employ a guidance scale of 12. The forward guidance of MEDIC is 1 while the reverse scale is 5. We chose these values by exploring different guidance scales, as discussed in Appendix D. We conduct all experiments in NVIDIA 4090. For our evaluation, we have selected the public pre-trained AudioLDM 2-Music model (Liu et al., 2023b). To ensure a thorough and multidimensional assessment, we measure performance using a suite of metrics that includes CLAP, LPAPS, Structure Distance, and FAD and conduct on an NVIDIA 4090. The computation of these metrics is facilitated by the CLAP models provided within the AudioLDM_eval package, which is publicly available at `https://github.com/haoheliu/audioldm_eval`.

Our methodology is aligned with the protocol established by Manor & Michaeli (2024), where we have adopted a forward guidance scale of 3 and a reverse guidance scale of 12 for DDPM inversion. In contrast, the DDIM inversion employs a guidance scale of 5, and SDEdit utilizes a guidance scale of 12. For Disentangled Inversion Control, we have determined the forward guidance to be 1 and the reverse scale to be 5. These specific guidance scale values are selected after extensive experimental exploration, the details of which are discussed in Appendix D.

### C.1    METRICS

**Objective Metrics** There are details about four metrics to evaluate the performance of our novel Disentangled Inversion Control framework: (1) **CLAP Score** (Elizalde et al., 2023): This criterion evaluates the degree to which the output conforms to the specified target prompt. (2) **Struture Distance** (Ju et al., 2023): Leveraging self-similarity of audio features to measure the structure distance between the source and edited audio. (3) **LPAPS** (Iashin & Rahtu, 2021; Paissan et al., 2023): An audio adaptation of the Learned Perceptual Image Patch Similarity (LPIPS) (Zhang et al., 2018), this measure evaluates the consistency of the edited audio with the source audio. (4) **FAD (Fréchet Audio Distance)** (Kilgour et al., 2018): Analogous to the FID used in image analysis, this metric calculates the distance between two distributions of audio signals.

**Subjective Metrics** To directly reflect the quality of the audio generated, we carry out MOS (Mean Opinion Score) tests. These tests involve scoring two aspects: MOS-Q, which assesses the edited quality of the audio, and MOS-P, which measures the content preservation of edited audio.

For assessing editing fidelity, the evaluators were specifically directed to "Does the natural language description align with the audio?" They were provided with both the audio and its corresponding caption. They were then asked to give their subjective rating (MOS-Q) on a 20-100 Likert scale.

To assess essential content preservation, human evaluators were presented with source audio, target audio, source prompt, and target prompt. They were then asked to answer the question, "To what extent does the target audio retain the essential features of the source audio, such as melody, instrumentation, and overall style?" The raters had to select one of the options: "completely," "mostly," or "somewhat," using a 20-100 Likert scale for their response.

Our crowd-sourced subjective evaluation tests were conducted via Amazon Mechanical Turk where participants were paid $8 hourly.

| Guidance Scale | | Structure | Background Preservation | | | CLIP Similariy |
|---|---|---|---|---|---|---|
| Inverse | Forward | Distance$_{\times 10^3}$ ↓ | LPAPS↓ | FAD↓ | MSE$_{\times 10^5}$ ↓ | CLAP Score ↑ |
| 1 | 1 | 8.56 | 0.12 | 1.17 | 3.25 | 0.51 |
| 1 | 2.5 | 11.97 | 0.15 | 2.49 | 4.54 | 0.56 |
| 1 | 5 | 15.99 | 0.17 | 4.22 | 6.07 | 0.61 |
| 1 | 7.5 | 15.99 | 0.17 | 4.22 | 6.07 | 0.59 |
| 2.5 | 1 | 22.80 | 0.20 | 6.39 | 8.65 | 0.30 |
| 2.5 | 2.5 | 14.24 | 0.16 | 2.50 | 5.40 | 0.46 |
| 2.5 | 5 | 14.46 | 0.16 | 3.31 | 5.49 | 0.53 |
| 2.5 | 7.5 | 15.51 | 0.17 | 3.94 | 5.89 | 0.53 |
| 5 | 1 | 29.94 | 0.24 | 9.81 | 11.36 | 0.20 |
| 5 | 2.5 | 29.16 | 0.24 | 9.11 | 11.07 | 0.22 |
| 5 | 5 | 22.15 | 0.20 | 5.59 | 8.40 | 0.36 |
| 5 | 7.5 | 17.57 | 0.18 | 5.57 | 6.67 | 0.48 |
| 7.5 | 1 | 31.41 | 0.25 | 10.62 | 11.92 | 0.20 |
| 7.5 | 2.5 | 31.05 | 0.25 | 10.14 | 11.78 | 0.20 |
| 7.5 | 5 | 29.20 | 0.24 | 9.32 | 11.08 | 0.24 |
| 7.5 | 7.5 | 24.16 | 0.22 | 7.33 | 9.17 | 0.34 |

Table 6: Ablation Studies on Different Guidance Scale

| Method | Inference Time |
|---|---|
| AudioLDM 2 | 42.5s |
| MusicGen | 83.3s |
| SDEdit | 44.3s |
| DDIM Inversion | 81.6s |
| MusicMagus | 89.0s |
| DDPM-Friendly | 33.3s |
| MEDIC | 92.0s |

Table 7: Inference Time across different methods.

## D  QUANTITATIVE RESULTS

**Analyses on Different CFG Scale** The lack of systematic experiments that determine the optimal combination of guidance scales for achieving the best editing performance, and analysis of how these guidance scales affect the final consequence in both reconstruction and editing, we conduct this experiment to find the best scales.

**Inference Time** We compare the inference time of our method with baselines, and the results are compiled in Table 7. MEDIC achieves the comparative inference time with generation models and inversion techniques. We will make an attempt to reduce the inference time of zero-shot music editing in our future work.

## E  POTENTIAL NEGATIVE SOCIETAL IMPACTS

MEDIC may also lead to potential negative societal impacts that are worthy of consideration. If the data sample of the training model is not diverse enough or biased, the AI-generated music may be overly biased toward one style or element, limiting the diversity of the music and causing discrimination.MEDIC could be used to create fake audio content, such as faking someone's voice or creating fake musical compositions, posing the risk of fraud and impersonation. Hopefully, all these issues could be taken into consideration when taking the model for real use to avoid ethical issues.

## F  LIMITATIONS

In spite of the remarkable outcome of our method, due to the limitation of the generation model we used, we are incapable of instigating a profound change.

Due to the numerous steps it requires (T=200), the duration of computing distance is quite long. Thus, we will implement a more powerful text-to-music generation model to support better editing, while trying to use a consistency model or flow-matching model to achieve high-quality and fast music generation in future work. We will make an attempt to edit more interesting and complex music tasks in the future.

# G QUALITATIVE RESULTS

For each type in ZoME-Bench, We provide samples to observe the capability of MEDIC intuitively.

## G.1 CHANGE INSTRUMENT

In Figure 5, we show the capability of MEDIC to change the instrument. Here we edit the ground truth music piece with the source prompt "a live recording of ambient acoustic [guitar] music" and editing prompt "a live recording of ambient acoustic [violin] music". The difference in instruments can be observed in the Mel-spectrum.

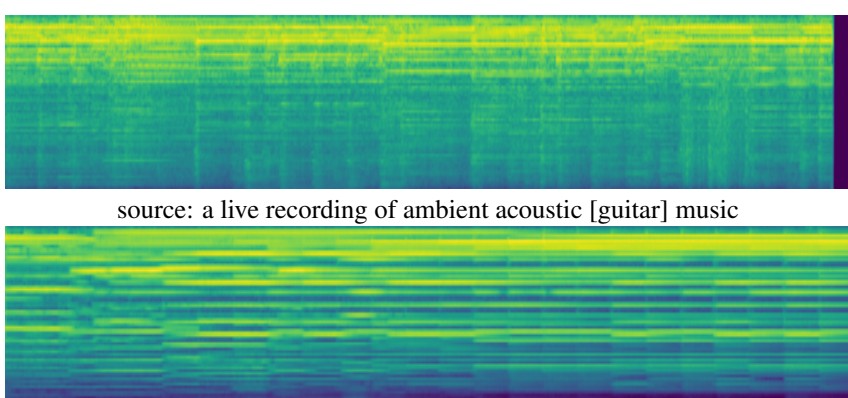

source: a live recording of ambient acoustic [guitar] music

edit: a live recording of ambient acoustic [violin] music

Figure 5: Editing Type 0 :Change Instrument

## G.2 ADD INSTRUMENT

In Figure 6, we show the capability of MEDIC to add more instruments. Here we edit the ground truth music piece with the source prompt "a heavy metal instructional audio with a distortion guitar" and editing prompt "a heavy metal instructional audio with a distortion guitar [and drums]". The appearance of the new instrument can be observed in the Mel-spectrum which presents a drum sound of high frequency.

## G.3 DELETE INSTRUMENT

In Figure 7, we show the capability of MEDIC to delete instruments. Here we edit the ground truth music piece with the source prompt "a lively ska instrumental featuring keyboard trumpets bass [and percussion] with a groovy mood" and the editing prompt "a lively ska instrumental featuring keyboard trumpets and bass with a groovy mood". The vanishing of the instrument can be observed in the Mel-spectrum.

## G.4 CHANGE GENRE

In Figure 8, we show the capability of MEDIC to change the genre of a music piece. Here we edit the ground truth music piece with the source prompt "a recording of a solo electric guitar playing [blues] licks" and the editing prompt "a recording of a solo electric guitar playing [rock] licks". The obvious difference in genre can be observed in the Mel-spectrum.

source: a heavy metal instructional audio with a distortion guitar

edit: a heavy metal instructional audio with a distortion guitar [and drums]

Figure 6: Editing Type 1 Add Instrument

source: a lively ska instrumental featuring keyboard trumpets bass [and percussion] with a groovy mood

edit: a lively ska instrumental featuring keyboard trumpets and bass with a groovy mood

Figure 7: Editing Type 2 Delete Instrument

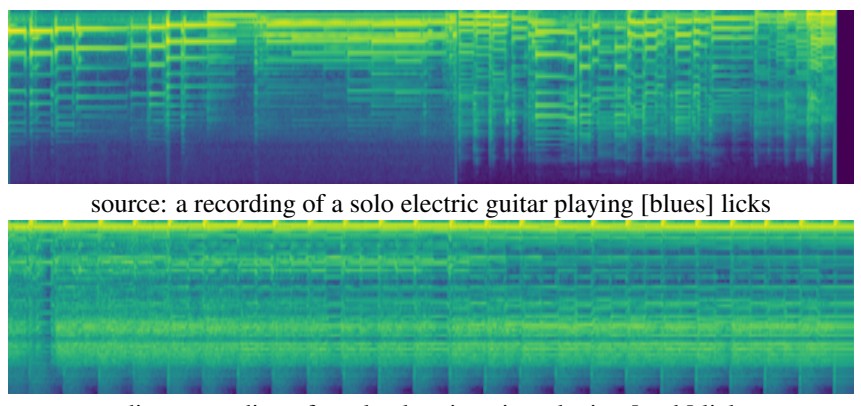

source: a recording of a solo electric guitar playing [blues] licks

edit: a recording of a solo electric guitar playing [rock] licks

Figure 8: Editing Type 3 Change Genre

### G.5 CHANGE MOOD

Mood is an important attribute of music. In Figure 9, we show the capability of MEDIC to change the mood of a music piece. Here we edit the ground truth music piece with the source prompt "a recording of [aggressive] electronic and video game music with synthesizer arrangements" and editing prompt "a recording of [peaceful] electronic and video game music with synthesizer arrangements". The change of mood can be observed in the Mel-spectrum.

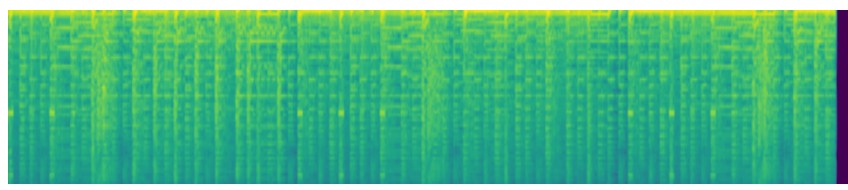

source: a recording of [aggressive] electronic and video game music with synthesizer arrangements

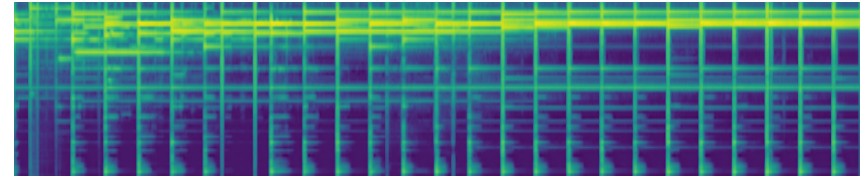

edit: a recording of [peaceful] electronic and video game music with synthesizer arrangements

Figure 9: Editing Type 4 Change Mood

### G.6 CHANGE RHYTHM

Rhythm represents the speed of the music. In Figure 10, we show the capability of MEDIC to change the Rhythm of a music piece. Here we edit the ground truth music piece with the source prompt "a [slow] tempo ukelele tuning recording with static" and the editing prompt "a [fast] tempo ukelele tuning recording with static". The change of Rhythm can be observed in the Mel-spectrum. The edited Mel-spectrum is much more intensive.

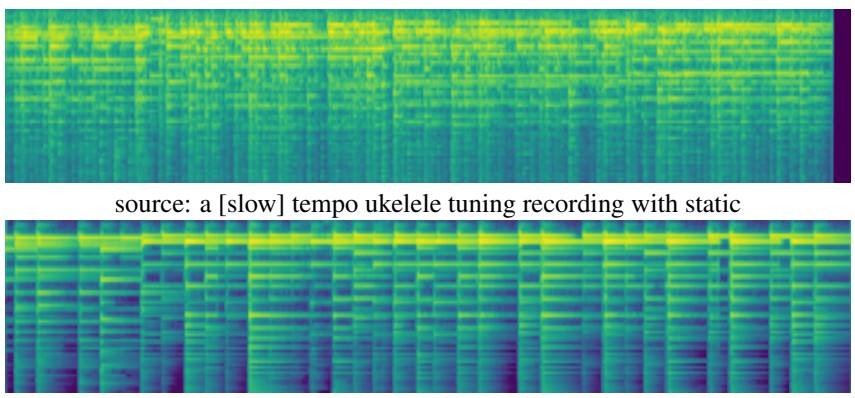

source: a [slow] tempo ukelele tuning recording with static

edit: a [fast] tempo ukelele tuning recording with static

Figure 10: Editing Type 5 Change Rhythm

### G.7 CHANGE BACKGROUND

In Figure 11, we show the capability of MEDIC to change the background of the instrument of a music piece. Here we edit the ground truth music piece with the source prompt "an amateur ukulele recording with a [medium to uptempo] pace" and editing prompt "an amateur ukulele recording with a [steady and rhythmic] pace". The change of instrument background can be observed in the Mel-spectrum.

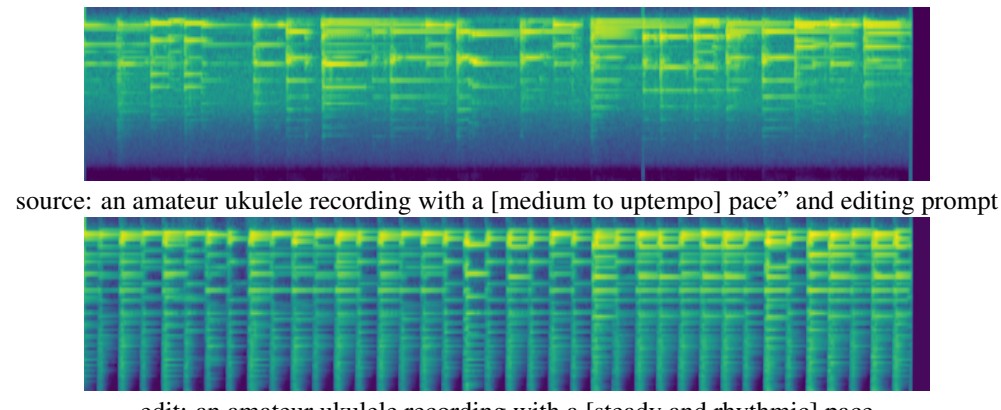

source: an amateur ukulele recording with a [medium to uptempo] pace" and editing prompt

edit: an amateur ukulele recording with a [steady and rhythmic] pace

Figure 11: Editing Type 6 Change Background

### G.8 CHANGE MELODY

In Figure 12, we show the capability of MEDIC to change the melody of a music piece. Here we edit the ground truth music piece with the source prompt "this instrumental song features a [relaxing] melody with a country feel accompanied by a guitar piano simple percussion and bass in a slow tempo" and editing prompt "this instrumental song features a [cheerful] melody with a country feel accompanied by a guitar piano simple percussion and bass in a slow tempo". The change of Melody can be observed in the Mel-spectrum.

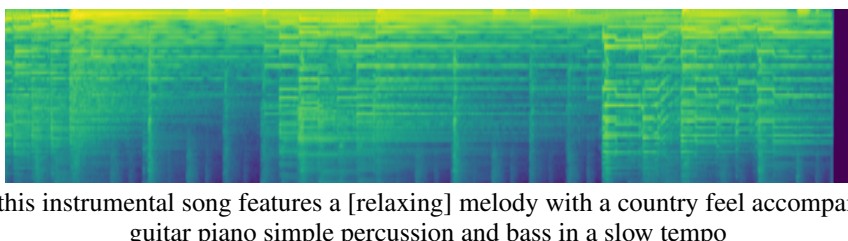

source: this instrumental song features a [relaxing] melody with a country feel accompanied by a guitar piano simple percussion and bass in a slow tempo

edit: this instrumental song features a [cheerful] melody with a country feel accompanied by a guitar piano simple percussion and bass in a slow tempo

Figure 12: Editing Type 7 Change Melody

### G.9 EXTRACT INSTRUMENT

In Figure 13, we show the capability of MEDIC to extract one certain instrument of a music piece. Here we edit the ground truth music piece with the source prompt "a reggae rhythm recording with bongos [djembe congas acoustic drums and electric guitar]" and editing prompt "a reggae rhythm recording with bongos". The change of instruments can be observed in the Mel-spectrum.

### H SAFEGUARDS

In the processing of the data and models involved in this study, we fully considered the potential risks. We ensure that all data sources are rigorously screened and vetted, and the model we used is absolutely trained from the safe dataset to minimize the security risks of being misused.

source: this instrumental song features a [relaxing] melody with a country feel accompanied by a guitar piano simple percussion and bass in a slow tempo

edit: this instrumental song features a [cheerful] melody with a country feel accompanied by a guitar piano simple percussion and bass in a slow tempo

Figure 13: Editing Type 8 Extract Instrument

