# OpenReview forum: "MEDIC: Zero-shot Music Editing with Disentangled Inversion Control"
_ICLR.cc/2025/Conference — ICLR 2025 Conference Withdrawn Submission_

### Official Review · Reviewer_ZuLk · 2024-10-30

**Soundness:** 2
**Presentation:** 2
**Contribution:** 2
**Rating:** 3
**Confidence:** 4

**Summary:**

This paper presents MEDIC, a novel method for training-free editing of musical audio with freeform text using pre-trained text-to-audio diffusion models. The paper also presents ZoME-Bench, a new editing benchmark for musical audio editing.

**Strengths:**

- The proposed method overall seems reasonably novel and well-motivated. Much space is given to explaining the facets of their method, and graphical comparison to existing methods like MusicMagus is very appreciated.
- Ablations of proposed method are solid and thorough, and shows clear strengths to the design choices the authors made.

**Weaknesses:**

Overall, while the proposed method is solidly novel and seems to perform better than current SOTA training-free editing approaches, issues in the overall clarity of the paper, evaluation suite, and in particular the proposed benchmark overweigh the contributions and thus I recommend rejection.



# Overall Clarity
---
The paper contains a number of grammatical errors, incorrect names of things, and incorrect citations.

- The “Branch” term (line 070) is introduced without explanation.
- "rigid" and "non-rigid" edits are introduced without explanation. These terms are not elucidated by figure 1, as there seems to be no difference between the input prompt and the "non-rigid" prompt.
- (Line 143) What is “plug-in-plus”? Is this a typo for plug-and-play?
- (Line 369) What is ZoMo-bench? Is this a typo from ZoME-Bench?
- It should be “MedleyDB” not “MelodyDB” (line 370)
- Appendix C seems to contain the same information duplicated twice (874-880 and 881-892)
- MusicCaps has the incorrect citation on line (161), which should point to Agostinelli et al.

While the above issues are minor, more importantly there is a distinct lack of clarity in the methodological contributions and what are contributions from the authors.
- In particular, it feels like the paper suffers from overuse of acronym-like terms. Between Disentangled Inversion Control, Harmonized Attention Control, and Disentangled Inversion Technique, it is hard to tell what is a subset of what.
- In section 3.3.1. even more acronym-like terms are introduced (Global Attention Refinement and Local Attention Blend), and it is unclear both A) why are these getting special names if they are from existing work where these terms are not used and B) if these are novel contributions, this is not made explicit.
- In Equation 2, there is a lone $M_t$, and it is unclear which attention map this refers to
- In equation 3/4, the threshold functions takes $w_{src/tgt}$ as argument but in equation 6 it is only a function of the mask and the threshold
- Both 3.3.1. and 3.3.2 are realtively hard to follow without an intimate knowledge of past (image-domain) works. These sections as a whole could be made more clear by drawing specific examples to editing tasks.
- 3.3.3 (and algorithm 2), it is a bit unclear whether “forward” refers to the forward diffusion process (i.e. data $\rightarrow$ noise) or the “forward” pass of the model. I think it is the latter, and if so I think this nomenclature should be fixed to the “backward” step of the diffusion process to bring it in line with standard diffusion terminology.
- Paragraph 2 continuously refers to some caption include “with noise”, but this does not exist anywhere
- What is MEDIC? It is sometimes used as (what I can infer) to be the main method, but this is never stated and it is unclear what this refers to specifically.

# Evaluation Suite
---
 - A key hyperparameter in all these editing-through-inversion methods is the $T_{start}$ parameter, which should in theory determine overall edit strength. It is unclear how this hyperparameter is chosen for the proposed method (and admittedly it may be ignored with using all 200 steps), but it is unclear how it was chosen for the baseline comparisons (such as DDPM-friendly).
- As DIC is a text-based editing method, it is unclear how the MusicDelta section of MedleyDB was used for the task. If the authors used the prompts generated from Manor and Michaeli, this should be explicitly stated. In general, it is unclear what sorts of edits this dataset even contains other than that the samples are longer.
- LPAPS results seem odd when comparing to Manor and Michaeli, as the values for table 2 are in theory on the same dataset, yet are all considerably lower (by about 10x) than the values reported in Manor and Michaeli. Similar inconsistencies hold for FAD, and in general it is unclear why the results for DDPM-friendly are different from the original paper (as supposedly this is the same data).
- Standard error bars should be reported for the subjective listening results, as all the values (with the exception of SDEdit) are quite close, and it is thus unclear whether the differences are statistically significant. In particular, it is also not stated how many subjects they used for the listening test and how many samples of each editing task were tested, as statistical significance should be calculated on average scores for each stimulus aggregated across users to assess whether such results would extrapolate to additional results from the proposed method.
- FAD is a rather odd metric to be using in this context with paired samples, both because A) it is an unpaired, distributional metric that ignores the structural relevance between pairs of samples and B) FAD has documented clear instability in small sample sizes for estimating the covariance matrix of each Gaussian (with generally needing 2.5k-5k+ samples to estimate this reliably [1]), thus making results somewhat unstable given ZoME-Bench’s size of only 1100 samples. Other metrics such as CLAP Maximum Mean Discrepancy (MMD) could be better suited, but in general it would make more sense to compare the FAD of generated samples to some existing reference set (such as Song Describer), as it is really only a measure of audio realism than anything on similarity to reference signals (which the text should reflect).
- The argument in the “Qualitative Results” section is reasonably heuristic. From visually inspecting the spectrograms, it is not clear of any structural relevance to the original source audio, and simply saying “the change of ___ can be seen in the Mel-spectrum” is insufficient to point to anything meaningful (though admittedly, the utility of spectrograms as a visualization here is not great). However, I think the overall success of the proposed method is somewhat overstated, as most of the examples provided in the **audio demo samples** do not actually perform the target edit and preserve the non edited content at the same time, with most samples sacrificing one of these two facets (which the authors identify as the most important parts of the editing process).

# Proposed Benchmark
---
My biggest issue with the paper is the proposed benchmark dataset of ZoME-Bench, as it seems to contain a number of logical flaws that severely limit its utility as a public benchmark.
-  It is odd that as the first “editing benchmark”, there is no actual ground truth information about the correctness of the edit itself. If it is only being assessed by CLAP score, this implicitly assumes that 1-2 word changes in the CLAP text prompt return meaningful changes in the input embedding in order to assess this change, which is assumed without support here. One could imagine here that in a truly suboptimal sense, a model could simply prioritize an edit that does absolutely nothing to the source audio but is able to change the CLAP embedding of the output, which would theoretically achieve perfect results on the benchmark. As the benchmark is a core contribution of the present work, the authors should either have ground truth target audio samples for each source audio (which would be easily doable for some of the instrument tasks if one had source-separated tracks), and/or at least followed the growing standard practice [2] in editing evaluation and use pretrained discriminative models to help assess the edit fidelity of more fine-grained tasks, which is fully doable in this case (such as using instrument tagging models for edits 0/1/2/8 or genre classifiers for edits 3/4).
- Many of the tasks are similar to previous work (AUDIT / InstructME) in being source separation tasks (0/1/2/8), and thus a much more natural choice for this benchmark would include source separated tracks in order to actually assess these edits that have real ground truth answers. While it is still unclear where the text captions came from for the MedleyDB subset (i.e. if they came from the Manor and Michaeli), it is odd that the MedleyDB subset was not used for creating the benchmark, as it seems readily available and has separated tracks for the instrument-based tasks, thus giving a possible avenue for ground truth targets.
-  The paper in particular is missing a rigorous definition of what “rigid” and “non-rigid” mean in the context of text-based audio editing, and why they have deemed certain editing tasks in one category or another. For the rest of my point here, I assume rigid means “content-based” and non-rigid means “style-based,” as that is what the paper seems to imply and is inline with past image domain works. For example, it is unclear why “instrument change” is referred to as a non-rigid task, given that in theory the change of an instrument should preserve all harmonic and melodic content and only reflect timbral changes (as it would be if a guitar player was changed to a violin but played the same part). Unlike in image domain tasks (where edits can mostly be grouped into those than edit a particular masked / bounding box region of the source image vs. ones that make global edits), this notion of region-concept cooccurrence does not exist in audio and thus porting over the definitions of “rigid” and “non-rigid” is not applicable out of the box.
- In general, I think a number of the tasks proposed do not make for an informative benchmark.  Tasks 3/4/5/6/7 (genre/mood/rhythm/background/melody) are ill-defined, as these conceptual buckets do not disentangle content vs. stylistic changes in the audio, and seem to be rather divorced from how actual musicians talk about semantic changes in music. As examples:
  - For genre, if “blues” changes to “rocks” then what changes? Is this a reflection of content changing (such as chords being simplified and melodic lines using fewer blues scales) or of stylistic changes (micro-timing, ornamentation, guitar fx)?
  - If “fast” is changed to “slow”, should the entire content be slowed down (thus reflecting a content-based change that can only be seen as “stylistic” if realignment is assessed between the original and now slower edit) or is this just a measure of decreased density of perceptual onsets?
  - If a “relaxing” melody is changed to a “cheerful” one, does this reflect changes in the pitch values, the rhythmic interpretation, both, or neither?
- While this is somewhat up to subjectivity, none of these semantic tasks seem non-rigid to me (as they all involve some amount of content preservation with stylistic changes). If these are meant to allow content-based changes, this should be explicitly stated, and in general, I’m hesitant to even phrase such hypothetical tasks as “edits” in the first place (as something has to stay the same for it to be considered an edit).


Between the issues with the non-rigid tasks, lack of ground truth for the rigid tasks, and over-reliance on CLAP similarity as a measure of edit accuracy, the overall use of this benchmark for standardizing and improving music editing is quite limited. To improve the paper, I think that either completely focusing on the MedleyDB subset and mostly dropping the proposed benchmark from the paper (as the methodological contributions stand on their own) or performing the significant work to improve the benchmark (and/or justifying why CLAP similarity can be used as a ground truth signal so heavily) would both be valid options, as the present version of the benchmark is my main concern.

[1] Jayasumana, Sadeep et al. “Rethinking FID: Towards a Better Evaluation Metric for Image Generation.” 2024 IEEE/CVF Conference on Computer Vision and Pattern Recognition (CVPR) (2023): 9307-9315.

[2] Basu, Samyadeep et al. “EditVal: Benchmarking Diffusion Based Text-Guided Image Editing Methods.” ArXiv abs/2310.02426 (2023): n. pag.

**Questions:**

Most of my questions are brought up throughout the previous section. In general, there are a number of what I think are typos (but I may be misunderstanding things), as well as my questions regarding the definitions of "rigid" and "non-rigid", which acronym-like terms refer to what, and questions regarding baseline reproduction and comparison.

---

### Official Review · Reviewer_PQrm · 2024-11-01

**Soundness:** 2
**Presentation:** 1
**Contribution:** 2
**Rating:** 3
**Confidence:** 5

**Summary:**

The authors propose MEDIC, a training-free music editing method utilizing pretrained diffusion models. MEDIC extends DDIM inversion to enable better music editing. Specifically, it achieves this by first obtaining the noise latent $z_{T}$ through standard DDIM forward sampling. During reverse sampling, MEDIC incorporates Cross-attention control, as proposed in Prompt-to-prompt, and Mutual Self-attention control, as proposed in MasaCtrl, while introducing "Harmonic Branch" for integrating Cross-attention control and Mutual Self-attention control.

Additionally, authors propose Disentangled Inversion Technique. This approach focuses on the difference between the latent $z^{*}_{t}$ obtained during DDIM forward sampling, and the source latent $z^{src}$, to guide the reverse sampling.

Alongside the MEDI, authors also introduce a new benchmark dataset, ZoME-Bench, designed specifically for music editing evaluation.

**Strengths:**

To improve the music editing performance of DDIM inversion, the authors did not simply combine Cross-attention control and Mutual self-attention control; they introduced an additional Harmonic Branch to integrate these techniques. Furthermore, they proposed the Disentangled Inversion Technique. By leveraging these methods, they surpass existing music-editing methods in both objective and subjective metrics.

Originality/Contribution:
- Introduction of the Harmonic Branch and Disentangled Inversion Technique for DDIM inversion
- Proposal of ZoME-Bench

**Weaknesses:**

**Overall**:

The following points represent the overall weaknesses in the current manuscript. Please refer to the detailed explanations in the latter part of the Weaknesses and Questions sections.
1. Insufficient or unclear validation of the effectiveness of the proposed method, which is directly related to the originality of this work. (For more details, see A. in Weaknesses and 1. in Questions.)
2. Unclear motivation for incorporating the inversion process (L3 in Algorithm 2) within the problem setup (where a source prompt $\mathcal{P}$ is provided). (See more details at 2. in Questions.)
3. The contribution of ZoME-Bench to the music-editing field seems somewhat limited. (For further details, see B in Weaknesses.)


**Details**:

A. The validity of the objective metrics used for evaluation remains unclear. For more details, please refer to Question 3.
- Given the ambiguity of these objective metrics, the experimental justification for the advantages of using the Harmonic Branch and introducing the Disentangled Inversion Technique seems insufficient.
- On the other hand, the benefits of combining Prompt-to-prompt and MasaCtrl appear to be adequately validated in subjective evaluation. However, this aspect alone may not be sufficient to fully support the originality and strengths of this work.

B. While I agree with the importance of introducing standardized benchmarks in audio/music editing and appreciate the effort to create a dataset, ZoME-Bench still has some limitations. ZoME-Bench includes original audio samples, original prompts, editing prompts/types/instructions, etc., but it lacks edited audio samples that define how the source samples are supposed to be edited in a certain sense. In this respect, although ZoME-Bench contributes to standardizing editing instructions, it leaves unresolved the larger issue of verifying the edited results, which remains a significant challenge in audio/music editing evaluation. Therefore, while ZoME-Bench contributes to the audio/music-editing benchmark to some extent, its impact is limited.
(I understand how difficult it is to construct such edited audio samples. I mention this point to assess and clarify the degree of the contribution and originality of the ZoME-Bench proposal.)

C. To improve the paper's presentation quality, I recommend revising the current manuscript. (See Comments in Questions.)

**Questions:**

**Questions**:

1. Objective Metrics

First and foremost, since evaluating audio/music editing tasks is not trivial, I think it is crucial to carefully select the evaluation metrics and design the evaluation protocol. On top of this, I have several questions about them.

- Regarding the FAD, LPAPS, and CLAP Score, could you specify which checkpoints were used to calculate each metric?
- For FAD, if it was calculated using VGG-ish, recent literature (such as in [1]) indicates that this model may not be appropriate for evaluating music samples. To support the effectiveness of the proposed method, I recommend using alternative pretrained models as suggested in [1].
   - For example, based on the correlation between MOS and FAD discussed in [1], it would be more appropriate to use FAD with LAION-CLAP to evaluate musical quality, and to use FAD with DAC/EnCodec embeddings to assess acoustic quality (please see more detail in [1]).
- For LPAPS, if the authors used the pretrained network from this repository [2], the network was trained on the VGGSound dataset [3] (also see its data filtering process). This raises some concerns regarding its validity for numerical evaluation in music editing. Additionally, the checkpoint provided in that repository [2] has other issues. As noted in this issue [4], the authors of this repository acknowledge problems in the training procedure of the LPAPS itself, and thus, they do not recommend using the LPAPS for at least training purposes.
    - It would be appropriate to calculate the L2 distance using other audio encoders trained properly. For instance, as in [1], I recommend calculating the L2 distance based on embeddings from audio encoders like LAION-CLAP, DAC, or EnCodec.
- Besides, could you provide at least an intuitive explanation, if not a theoretical one, supporting why LPAPS is suitable for evaluating "consistency"?
- CLAP model: Appendix C refers to this repository [5], but I couldn’t find the CLAP model there. Could you clarify this in more detail?


2. In cases where a source prompt $\mathcal{P}$ is provided, is there a benefit to using the inversion process in L3 in Algorithm 2? It seems that just using the proposed attention control technique in Section 3.3 during reverse sampling alone might be sufficient. From an ODE perspective, the score function at a given timestep should be almost the same in both forward and backward directions in terms of conditioning. The difference between them would be the accumulated errors from $z_{0}$ and $z_{T}$. If L3 were removed, it would no longer be 'inversion'. It would be 'text-guided music editing by attention map control' such as in MusicMagus.

3. The definitions of "rigid" and "non-rigid" tasks mentioned in the Introduction are unclear in the paper, leaving some doubts about the validity of claims regarding the proposed method’s effectiveness. Even the example provided around L321 in Section 3.3.3 does not seem intuitive enough. Could you elaborate more?

4. In Section 2, L143, the authors state, "Differently, we introduce a plug-in-plus method called Disentangled Inversion Control to separate branches, achieving superior performance with considerably fewer computational resources." Was this claim tested thoroughly in the paper? From Table 7, the computational cost of the proposed method appears to be higher than that of the baseline methods (also, it seems that Null-Text Inversion is not included as a baseline).

**Comments**:
- In diffusion model literature, the terms ‘forward process’ and ‘backward process’ typically refer to the process from $z_{0}$ to $z_{T}$ and $z_{T}$ to $z_{0}$, respectively, even when dealing with inversions [6][7]. To minimize unnecessary confusion for readers, I recommend revising the current manuscript to maintain consistency with prior work in fundamental aspects. (In fact, in Appendix C, the authors use the term “forward guidance” naturally.)
- In Figure 1, citations to MusicMagus should be included (for a self-contained perspective). There are instances of subjective terms like “tiny distance,” “small distance,” and “large distance” without clarification on what these distances pertain to. While the intent becomes clearer upon multiple readings, I suggest revisions to improve clarity, allowing readers to grasp the meaning on the first read-through. Additionally, the term "two-branch inversion techniques" does not appear to be a widely recognized term in inversion, I feel.
- Missing explanations for indices $i, j, k$ in Section 3.3.1. Also, Eq (6) is not consistent with Eq (3), (4).
- The values of hyperparameters such as $S, L, \tau$ are not explained in the experiment section.
- Section 3.4, L356–L358: Citing only image-editing literature while discussing audio/music editing seems wired.
- In Appendix C, content from L881 is repeated.

[1] Gui, A., Gamper, H., Braun, S. and Emmanouilidou, D., 2024, April. Adapting frechet audio distance for generative music evaluation. In ICASSP 2024-2024 IEEE International Conference on Acoustics, Speech and Signal Processing (ICASSP) (pp. 1331-1335). IEEE.

[2] https://github.com/v-iashin/SpecVQGAN

[3] Chen, H., Xie, W., Vedaldi, A. and Zisserman, A., 2020, May. Vggsound: A large-scale audio-visual dataset. In ICASSP 2020-2020 IEEE International Conference on Acoustics, Speech and Signal Processing (ICASSP) (pp. 721-725). IEEE.

[4] https://github.com/v-iashin/SpecVQGAN/issues/13

[5] https://github.com/haoheliu/audioldm_eval

[6] Song, J., Meng, C. and Ermon, S., 2020. Denoising diffusion implicit models. ICLR 2021

[7] Parmar, G., Kumar Singh, K., Zhang, R., Li, Y., Lu, J. and Zhu, J.Y., 2023, July. Zero-shot image-to-image translation. In ACM SIGGRAPH 2023 Conference Proceedings (pp. 1-11).

---

### Official Review · Reviewer_AcUQ · 2024-11-04

**Soundness:** 3
**Presentation:** 2
**Contribution:** 2
**Rating:** 5
**Confidence:** 5

**Summary:**

This paper primarily discusses a method for enhancing the performance of zero-shot music audio editing tasks through multiple control mechanisms, referred to by the authors as Disentangled Inversion Control. Additionally, the paper contributes a benchmarking dataset based on MusicCaps, aimed at evaluating the performance of zero-shot music editing models.

**Strengths:**

- The main idea of this paper—incorporating mutual self-attention, cross-attention control, and harmonic control—is sensible, even though each module is not entirely novel. The combination of these mechanisms appears effective, as results indicate that combining them enhances model performance in music editing tasks, providing useful insights.

- The paper is thorough in its experimental design, including both subjective evaluations and a variety of objective experiments. The results effectively demonstrate the validity of the chosen methods for the model.

- The discussion of related work is comprehensive.

**Weaknesses:**

Although this paper is a strong empirically-driven study, there are certain hypothesis-related issues that could be improved.

- First, the paper needs to clarify what is meant by “rigid” and “non-rigid” tasks. These terms appear throughout the paper, but after re-reading the entire text, I still found no clear explanation of what these tasks entail, which left me quite confused.

- The paper actually addresses a text-guided music audio editing task. However, the language and context in the main body do not consistently maintain this focus. Given the current context, I suggest aligning terms in the main text to match the title, shifting from “audio editing” to “music editing.”

- While the proposed multiple control method indeed focuses on different aspects through each control mechanism, whether this approach achieves “disentangled” control is debatable. To demonstrate that the controls are disentangled, the paper should include experiments showing that one control does not interfere with another. While these controls focus on different levels conceptually, they do not intuitively seem orthogonal, making the term “disentangled” potentially misleading. I suggest either adding experiments to confirm this or revising the terminology.

- The paper includes a subjective evaluation, which is commendable. However, the description of this evaluation is incomplete. Typically, subjective evaluations should also describe the gender, age, music background, and musical training distribution of the subjects, which helps with the interpretation of the results. Unlike data annotation, where these factors might be less crucial, they are important here due to potential biases introduced by AMT, and these underlying biases should be considered.

- In addition, hypothesis tests should be conducted for all reported results.

**Questions:**

- The benchmark dataset proposed in the paper is a good idea, but upon reviewing it, I found that it only includes a single audio file. Could the authors further clarify what constitutes the ground truth in this context?

- Finally, I am very curious about the computational efficiency of this method. Does it require more time and resources compared to baseline methods?

---

### Official Review · Reviewer_DvZA · 2024-11-04

**Soundness:** 3
**Presentation:** 1
**Contribution:** 3
**Rating:** 5
**Confidence:** 3

**Summary:**

This paper propose a new approach to do zero-shot music editing by Disentangled Inversion Control, which integrates multiple methods to inject the diffusion process. A novel benchmark is also proposed.

**Strengths:**

1. Good zero-shot editing performance compared to previous STOA. The demo page shows the effective controllability of some music concepts that previous models failed to control.
2. The benchmark is very useful for future researchers on music editing.
3. The methodology of Harmonized Attention Control and Disentangled Inversion Technique is novel, which could help zero-shot editing of other domains.

**Weaknesses:**

1. While the experiments are about music editing, the evaluation only uses metrics for general audio editing. Music content-related metrics like chroma distance [1] are missing.
2. The paper does not seem to be clear enough. See questions.
3. The values and effects of the hyperparameters in the paper are unclear, like $k, \tau_c, L$ and $S$. Ablation study or case study by changing these hyperparameters would be helpful to understand the model.
4. While the methodology seems to be general-purposed, the experiments only focus on music editing. This is okay but limits the impact of the paper a bit.
5. Typos and formatting errors. Algorithm 1: Inconsistency use of fonts; formatting error of \hat in $\hat{M}^{\textrm{tgt}}$; $\epsilon_{c_{\textrm{tgt}}}$ should be $\epsilon_{\textrm{tgt}}$. Figure 2: inconsistenct notation $M_{\textrm{tgt}}$ vs. notation in text $M^{\textrm{tgt}}$. Equation 9: $l$ is not defined. Table 6: not referenced in the appendix text.

[1] Zhang, Y., Ikemiya, Y., Xia, G., Murata, N., Martínez-Ramírez, M. A., Liao, W. H., ... & Dixon, S. (2024). Musicmagus: Zero-shot text-to-music editing via diffusion models. arXiv preprint arXiv:2402.06178.

**Questions:**

The introduction and methodology has some places that are unclear to me.

1. What is rigid and non-rigid editing in the context of music editing?
2. In 3.3.1 Global Attention Refinement, equation 2, should the second case be $(M_t)\_{i,A(j)}$ instead of $(M_t)\_{i,j}$?
3. In 3.3.1 Local Attention Blends, the definition of $\textrm{Threshold}(\cdot,k)$ does not match the format in equation 3. Also, what are the choices of $k$ and how will different $k$ affect the results?
4. In 3.3.1 Scheduling Cross-Attention Control, the usage of $\textrm{Refine}(\cdot,\cdot)$ does not match the definition in equation 2.
5. In 3.3.1 Scheduling Cross-Attention Control, what is the choice of $\tau_c$ and how will it affect the results?
6. In figure 2, the harmonic branch outputs something that looks like a "rapid guitar music." Is it an observable phenomenon in experiments, or is it just an assumption? Does the upper part handle non-rigid editing and the lower part handle rigid editing only?
7. In table 4, why are there no results for [0, 0, 0]?

---

### Note · Authors · 2024-11-20

I have read and agree with the venue's withdrawal policy on behalf of myself and my co-authors.